# Water Value Integrated Approach: A Systematic Literature Review

Jean de Paula [1,2,*] and Rui Marques [2]

1   Institute for Applied Economic Research (Ipea), Setor de Edifícios Públicos Sul 702/902, Bloco C-Torre B, Asa Sul, Brasília 70390-025, DF, Brazil
2   Civil Engineering Research and Innovation for Sustainability (CERIS), Instituto Superior Técnico, Universidade de Lisboa, Av. Rovisco Pais, 1049-001 Lisbon, Portugal; rui.marques@tecnico.ulisboa.pt
*   Correspondence: jean.marlo@ipea.gov.br

**Abstract:** Extreme water incidents point out a value conflict surrounding the resource. While drought and floods echo the inadequate land and resource uses, the increase in social inequalities exposes the practical, physiological, and social consequences. The multiple value action throughout the water cycle also narrowed disputes to those that neglect its vital importance, and the constraints imposed to the services, such as low tariffs, and lack of local engagement, make sustainable water systems more difficult. This article develops a systematic literature review to understand the academic motivations surrounding water value and gaps in its systematic approach. A sample with 84 papers is created by an interactive keyword selection and its general characteristics are presented. A dynamic reading technique extracts data and classifies the papers according to 14 research motivations, where the water multifunctionality and the user value stand out. The bibliographic coupling analysis identifies a cluster of 16 papers related to integration and connected to planning, decision, and management. There is a lack of contribution with a systemic approach to water resources by way of integrating actors and values, such as including local contexts.

**Keywords:** water values; system approach; integrated water approach; systematic literature review



## 1. Introduction

Incidents throughout the water cycle raise the issue of a conflict of values with extreme practical, physiological, and social consequences. Floods reflect the discrepancy between inadequate land use and water runoff. Scarcity and lack of access to drinking water compromise the most basic human needs, increasing social inequalities and conflicts that take progressively violent forms in various countries. Traditional sectoral discussions focus on the impact of climate change on energy generation, transport, and food production, dismissing the need to restore rivers, underground reservoirs, soil, and related ecosystems.

Faced with these and other disputes, there is a need to emphasize water's basic values for society and to guide efforts in this direction. In 2002, the United Nations (UN) declared water (and sanitation) as a mandatory precondition for human rights [1], giving it a specific and central chapter in the Sustainable Development Goals (SDGs). The SDGs align a multidisciplinary challenge to human rights through environmental, economic, and social priorities.

Drinking water is addressed by goal 6, broken down into eight indicators to highlight the need for an integrated approach and financing. Griggs et al. [2] illustrated the resource's synergy with hunger and marine life (SDGs 2 and 7) by its dependence on food and nutrients. The goal report [1] mentions its interdependencies with cities (SDG 11), production and consumption (SDG 12), land use (SDG 15), and partnerships (SDG 17). SDG also emphasized its integration throughout the water cycle, namely with a joint concern for preservation, quality, recycling, and use of the resource, recognizing it as an important theme for its governance.

On financial issues, UN-Water [1] remarks that the sector's nature requires special resources and attention because its landscape integration and the complexities of the assets, such as scale, maturation time, and spatial intervention, are not often noticed in contemporary life, as Frischmann [3] commented. Lankao [4] used the examples of Mexico City and Buenos Aires to illustrate the sectorial challenges illustrated via four problems: insufficient investments, financing difficulties, political interference, and prioritization of new projects. In other words, the author's findings suggested that the political return achieved by lower fares resulted in insufficient revenue for reinvestment, further compromising operational efficiency, which discourages new investors. This logic is outside the political equation since the public resources guarantee the visibility of new projects.

Asset deterioration as a result of a lack of financial resources, undermines the operation, maintenance, and use-values, imposing exponential interruptions to services over the infrastructure's life cycle. At the same time, the vulnerability of these services increases the collective losses to the system itself and other related ones, which implies overall losses greater than individual gains in tackling "inefficiencies". The consolidation of urban spaces requires more complex intervention technologies to avoid downtime in the local routine. Kim et al. [5] commented that the increasing occurrence of failures in the water supply system in South Korea resulted from the increasing demand on assets at the end of their useful life. The authors predicted the deterioration of 33% of these networks in that country in the 2020s, requiring over Euro 13 billion (KRW 18 trillion) to ensure water supply. Mazumder et al. [6] pointed out that a burst pipe in a water network has a greater impact on the performance of local roads (9.6% reduction) than on the water network itself (7.5%). If, on the one hand, pursuing efficiency can be justified by resource preservation, on the other hand, it may impose limitations on the beneficiaries if it subjects them to a sector with difficulties in developing or improving technologies and services.

Thus, the relativity of values becomes a central aspect of the discussion given the polarization that arises from these clashes. Van Gestel et al. [7] identified those public values, such as economic development and environmental quality, which are being diluted by other particular values that emerge later, such as efficiency, transparency, and legitimacy, and their relevance is still questioned at the end of the process. For this reason, to be explicit, the values at stake and their causalities may clarify the discussion in different contexts and promote greater consensus by signaling the interdependencies between collective and relative motivations. Such delimitation may also corroborate to measure progress and its outcomes in order to qualify the decision-making processes, balance values, and resources, and mitigate constraints in the system caused by the mutual motivations.

This article develops a systematic literature review to identify the motivations of related studies and then highlight the main water values focused on by academics, such as business performance or its conservation, health, nutrition, and human development. It develops an open methodology using bibliometric data and visual analysis that suggest thematic affinities, point to key institutions and countries interested in this theme, and uses a comprehensive presentation to deal with the sectorial characteristic of multi-level and -disciplinary stakeholders. In due course, it describes characteristics, such as the author, journals, and affiliations' prominences, as well as the chronological publication progress, countries of study, and linked themes. In addition to this introduction, Section 2 develops a brief literature review to highlight important concepts. Section 3 presents the study method with criteria and search tools used, while Section 4 describes the results found. Section 5 brings together the main narratives of the selected papers and Section 6 concludes the study.

## 2. Basic and Relative Values

The interests that clash around the resource show a polarity of values that compromise its sustainability. Schwartz and Bilsky [8] defined values as goals or objectives that influence preferences and justify initiatives to explain decisions, attitudes, and behavior. Particular, or individual, values arise, for example, from the innovations and self-development interest, in

contrast to collective values that spur social organization by conformities and universalism, among others.

Sagiv et al. [9] suggested that the study of values in psychology promotes a consolidated construct by reflections on the convergence of the needs of an individual or collective motivations to achieve a certain goal—shifting the interaction between agents from confrontation to conflict. Similarly, the predominance of certain motivations in the water literature may suggest its appropriation by particular or collective values, such as the cited imbalance between land misuse or sectoral concerns and housing security or soil nutrient recomposition.

Gondhalekar et al. [10] exposed the echoes of the collective and private water values clashes by ratifying, respectively, the direct relationship between diarrhea and access to safe drinking water and assigning some share of blame to tourism for the pressure of fluctuating demand in the Leh District, India. Water is also responsible for life quality through power generation [11,12], agricultural production [13–16], land development [17], security [18], and mitigation of social conflicts [19,20]. They also related it to the satisfaction of specific groups that seek business development through higher enterprises' performance [6,21–23]. Such motivations, or goals, expose the predominant value attributed to water—be it a business, human survival, or environmental conservation.

According to Torres et al. [24] and Sagiv et al. [9], part of the clashes between values stems from disagreements caused by the generic and self-referential appropriation of the term. Koppenjan et al. [25], for example, showed that the literature questions the public value as a universal good in the face of a growing clash of interests and a variety of definitions of the term, approaching the discussion in the classical and relative forms. The classical perspective sets the collective values as a public authority obligation to guarantee inalienable, solid, and objective human rights—a contradictory approach that requires further investigation, according to the authors. The relative perspectives have questioned these collective values in the face of the particular and institutional interests, citing those positive interventions for the society to meet local resistance by changes in individual contexts or by the characteristics of the actors involved (namely habits, rationalities, and structural and cultural environments). Thus, the recognition of the resource vitality is compelled by the particular interest's projection.

These contrasts are clear in the preservation of resources and dominion (or even the "backyards") to the detriment of diffuse benefits, reproducing conflicts between individual and collective values. In the first case, Schwartz and Bilsky [8] understood that new ideas, initiatives, and experiences promote self-development, unlike the collective interests related to conservative values with rigid conformities.

Disagreements also occur between individuals or groups. Bauke Steenhuisen and Michel van Eeten [26] highlighted the infrastructure operators' difficulties with the multiple principles arising from divergent demands. Political instances are also peculiar contexts in this process where problems, solutions, and policies result in government agendas "not necessarily with logical criteria such as its importance or criticality; but associated with issues such as interpretation, receptivity of the issues, the consistent dissemination of certain problem or solution and performing the political forces involved" [27]. Such criteria are legitimized both by voters and public agents who jeopardize long-term objectives to perform local services, or even guarantee their own interests, corroborating with the promotion of mere expectations.

Schwartz and Sagie [28] discussed the inefficiencies generated by confrontations, correlating values with socioeconomic development and democracy. As a premise, the authors considered that consensus is the basis for group harmony by promoting cooperation and mutual recognition, discouraging conflicts, and strengthening the acceptance of norms for the achievement of common objectives. The study confirmed the positive correlation between the importance of individual values (openness to experiences) and collective values (self-transcendence), but they were surprised by the contradiction arising from the context of these values. For the authors, low universal consensus in less developed

countries stems from greater survival needs—leaving, thus, the greater recognition of this universality when these basic needs are satisfied.

The relativity of values found by Schwartz and Sagie [28] referred to the importance of the theory's major principles. For Schwartz [29], these were basic values for decision-making that result from biological and social needs—namely interaction, survival, and well-being. If water is an inalienable good that objectively expresses its importance for the survival of organisms, the dissent over universalization in less developed countries exposes the subjectivity or relativeness of the values.

By presenting a hierarchy of human needs, Maslow [30] highlighted they assume a dynamic character because of the various combinations and forms. The satisfactions at mealtimes are an author example of this dynamic: in many cases, the status interests outweigh the importance of the physiological needs' satisfaction. Schwartz and Bilsky [8] exemplified the value of relativity by commenting that individuals' biological need for reproduction can be interpreted as the value of intimacy, a requirement for strengthening equity, harmony, and national security. Thus, the subjectivity of human needs results in more complex values that arise from group coexistence and may relativize biological needs. Therefore, the doubts of Koppenjan et al. [25] about classical public values initially caused strangeness because of questioning basic needs for human development, but they could be expressing the complexity of the organic perspective of these principles.

## 3. Method

This research has as its reference the systematic review and meta-analysis protocols proposed by Shamseer et al. [31] (Preferred reporting items for systematic review and meta-analysis protocols—PRISMA-P), notably in 12 items (17 considering sub-items) listed in the introductory and method sections. As presented by Lima et al. [32] and Yu et al. [33], the progressive selection of articles follows five stages: planning, collection (including search engine and keywords selections), treatment, and data analysis and presentation. The Figure 1 summarizes and underlines key activities per stage.

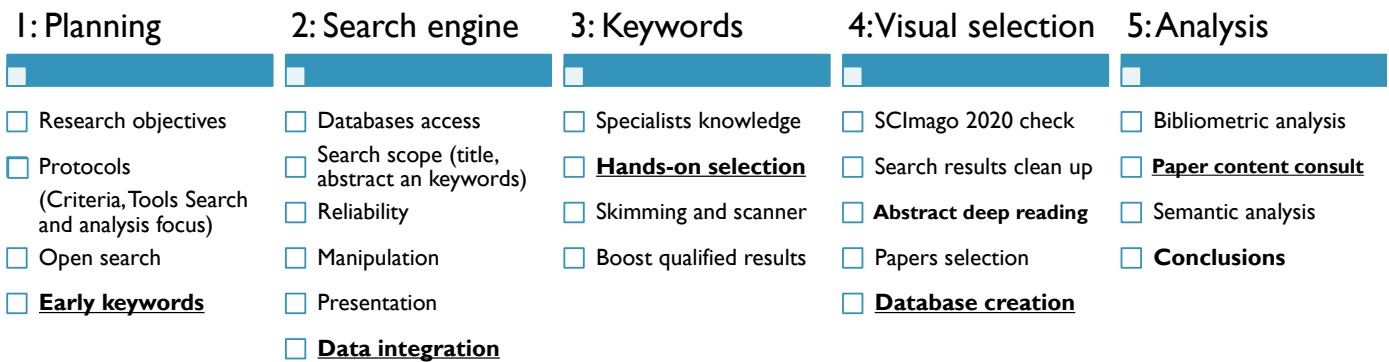

**Figure 1.** Steps methodology diagram.

Stage 1 defines the research strategy, pointing to the objectives and the protocols to be followed. This process begins with the delimitation of the research aim, giving the authors' knowledge, experts' discussion, and extensive reading of academic and grey literature found on Google and Google Scholar search sites. Next, the article selection criteria, the search tools, and the analysis focus are established, according to the resources and desired results. The relevance, objectives and context of the research are part of this article's introduction, while the following stages describe the eligibility criteria.

This initial effort (stage 1) provides the preliminary keywords ("valu *", "sanitation", "infrastructure") to select the search tool (stage 2), which received particular attention considering the fast development of search tools. Such selection depends on the planned approach, namely the search comprehensiveness and scope in the structure of the articles, trustworthiness, and reliability, as well as the manipulation, presentation, and integration of the data with each other and other sources and tools.

Gusenbauer and Haddaway [34] established 27 criteria to compare 28 search tools, among which they recommended half for a systematic literature review. These data were double-checked in the present study to maximize the search results, identifying 21 satisfactory search tools, according to 7 criteria. Three of them (BASE, Scopus, and Web of Science: Web) were selected for comparison considering the size of the databases, together with the Lens.org tool, considering its recent improvements, available functionalities, and coverage of other databases. In view of the reduced number of researchers and time, Scopus was selected because of the reduced data missing (years, abstracts, authors, keywords, nationality of authors, sub-areas, and journals, among others), no repetition, and integration with other data (Scimago Journal & Country Rank) and analysis tools (Excel, eulerAPE and VOSviewer).

The articles search continued with the keywords refining (stage 3) for a specific period (between 2011 and 2021) and language (English). Despite the experts' contributions, an interactive process is required to maximize and qualify the results with the alternation of words and linguistic structures. Successive adjustments are made until similar results are found, exploring a content analysis of the results with dynamic reading techniques commented on by Maxwell [35].

This stage shows the research complexity as the use of simple keywords results in a high number of answers in several areas of interest. There is a diversity of terms used in this context, making it necessary to <u>alternate the search between the most recurrent ones</u> and using compound words with up to 3 intermediate fragments. Figure 2 presents this refinement progress with each inclusion of compound words in the titles, abstracts, and keywords of articles published between 2011 and 2021, and summarize the results found by the code TITLE-ABS-KEY ((valu * OR success * OR benefit OR impact) AND ((infrastructure OR service OR utility) W/3 (water)) AND ((system * OR integrated OR sustain *) W/3 (approach OR plan * OR management OR evaluation)) AND PUBYEAR > 2010 AND PUBYEAR < 2022 AND (LIMIT-TO (DOCTYPE, "ar")) AND (LIMIT-TO (SUBJAREA, "SOCI") OR LIMIT-TO (SUBJAREA, "BUSI") OR LIMIT-TO (SUBJAREA, "ECON"))).

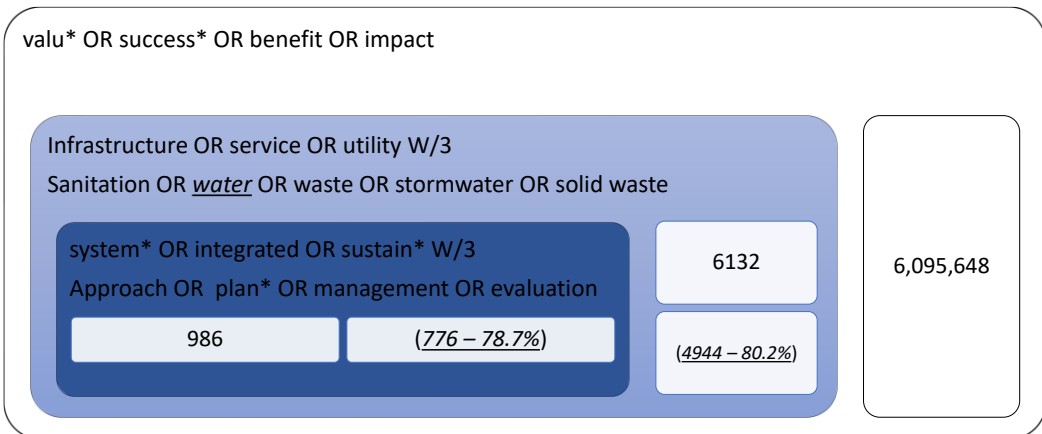

**Figure 2.** Search results per keyword block, highlighting the results focusing on the word "water".

The word "water" is prioritized due to its predominance among the others (sanitation, waste, stormwater, and solid waste), as well as the social and economic themes (including business, sample "water SE"–aUb) over the environmental one because of the diversity of intertwined disciplines. The representation of these three groups (SOC, ECO, and ENV) in Figure 3 further shows that the environmental issues still influence the sample water SE—illustrated by the overlapping circle "c" over areas "a" and "b".

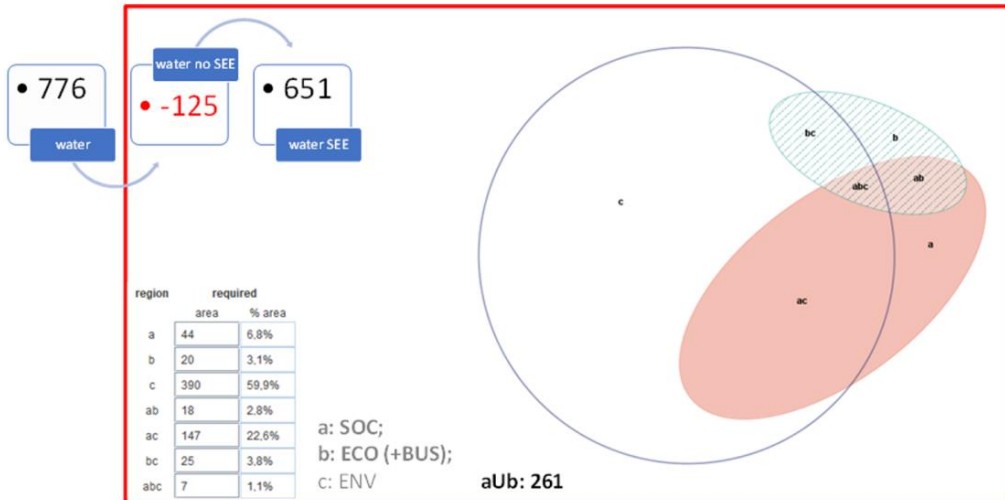

**Figure 3.** Dimensions in the research universe highlighting the "water SE" sample. Elaborated using eulerAPE−https://www.eulerdiagrams.org/eulerAPE (accessed on 6 January 2022).

The visual analysis of the results (stage 4) seeks to identify double entries, errors, or null values. None of these noises are identified, but six are excluded because they were published in journals that are not classified in Scimago 2020. In the remaining 255 qualified papers, the abstracts were analyzed to establish the study sample (84) for this research theme. When necessary, a double-check reviews the selected papers to extract and tabulate complementary data.

The quali-quantitative analysis (stage 5) developed a sample bibliometric and semantic analysis, as well as presents the key issues related to the research theme by an contend analysis. Several interests of a paper were framed as multiple uses of water or even performance issues as operational. The way in which they were addressed, such as an operational improvement for business, user or environmental benefits, or an explicit mention, highlights an integrated approach.

The prioritization of results is progressive and depends on the relevance that data adds to this research. For instance, the scale of analysis (countries of interest) is prioritized over a basic description of the nationality of institutions. The data are then synthesized through graphs and maps of network to clarify, for example, the distributions and linkages between the selected papers, as well as the relevance of authors through total citations or other information available.

## 4. Quantitative Review

### 4.1. Time and Subject Distribution

The water SE sample was broken down into not related (n/r) and selected articles to compare it with the environmental theme (ENV). As Figure 4 illustrates, during the analysis period there was an overall increase in publications, but the ENV theme has remained constant since 2014. This behavior showed a real growth of social and economic water interest (water SE sample), captured by the study sample since 2018 when 70% of the selected papers were published.

The stratification of the themes remaining in the study sample indicates the water multi-disciplinarity, presented in Figure 5. The predominance of the social theme (31.6%) was noticeable and the mentioned environmental influence (26.3%) was confirmed. The theme ECO (+BUS) (9.7%), in its turn, highlighted a slight advantage over energy (9.2%) and engineering (8.8%).

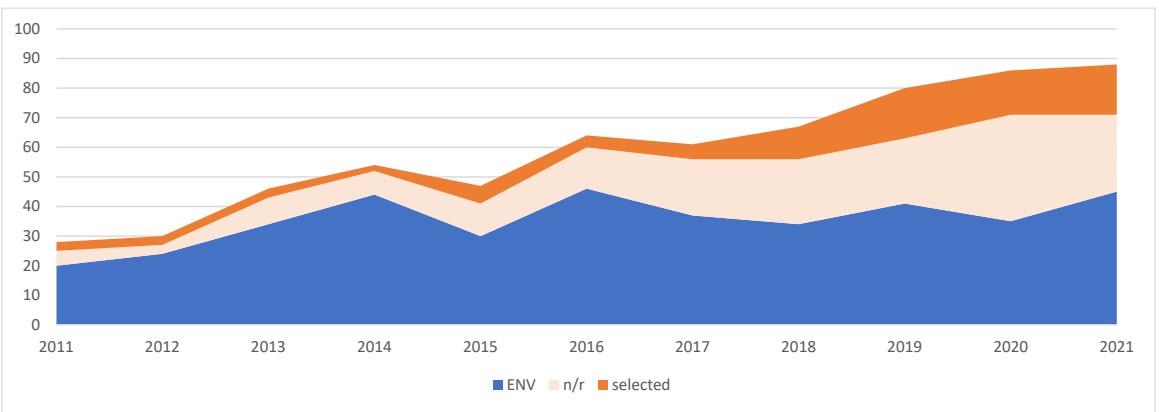

**Figure 4.** Number of publications per year of the "water SEE" sample. Adapted from Scopus.

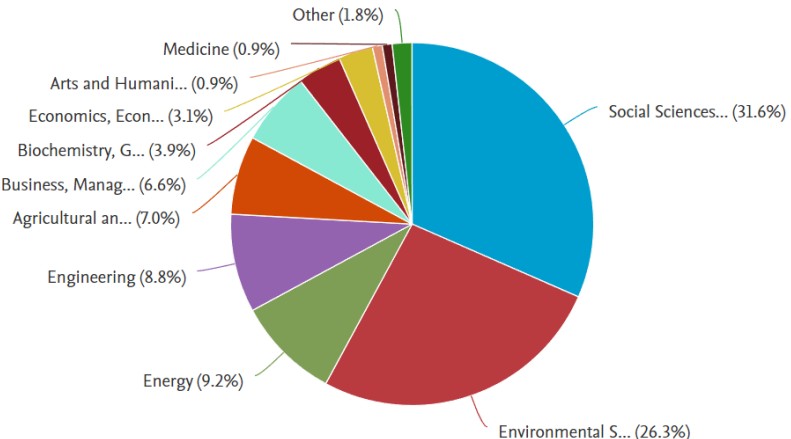

**Figure 5.** Number of documents per theme. Source: Scopus.

*4.2. Journals Overview*

The 84 selected articles were published in 47 journals, resulting in less than 2 articles per journal (1.8). Despite this diversity of sources, the study sample was categorized according to the articles number, total citations, and most cited articles to identify the popular journals. As underlined in Figure 6, 5 are regularly listed in the top 10 (Water Switzerland, Journal of Cleaner Production, Environmental Science and Policy, Ecosystem Services and Landscape Ecology). They have a common interest in environmental science thought themes, such as production, policies, and management. Water is the specific focus of one of them, covered by 13 research areas, as well as the economic and social values of ecosystem services.

The ranking progress of these five journals is shown to the left of Figure 7, illustrating the similar performance of most of them over time with increasing SJR rankings since 1999—even those more recently published (Ecosystem Services and Water Switzerland). The journal Landscape Ecology diverges from this behavior given the drop in ranking since 2010. It resumed its ranking as at its beginning (1999 and 2000). Still, these five journals are positioned in the first quartile of the rank Scimago 2020 (SJR > 0.6), suggesting well-qualified content of the sample.

To the right of Figure 7, box plot graphics of the article's classification present three views: general (SJR), water SE, and study samples. The criteria chosen for this research resulted in the progressive selection of better-qualified journals, given the greater concentration in the upper quartiles, and this result is consistent with the greater quantity of selected articles published in Q1 qualified journals.

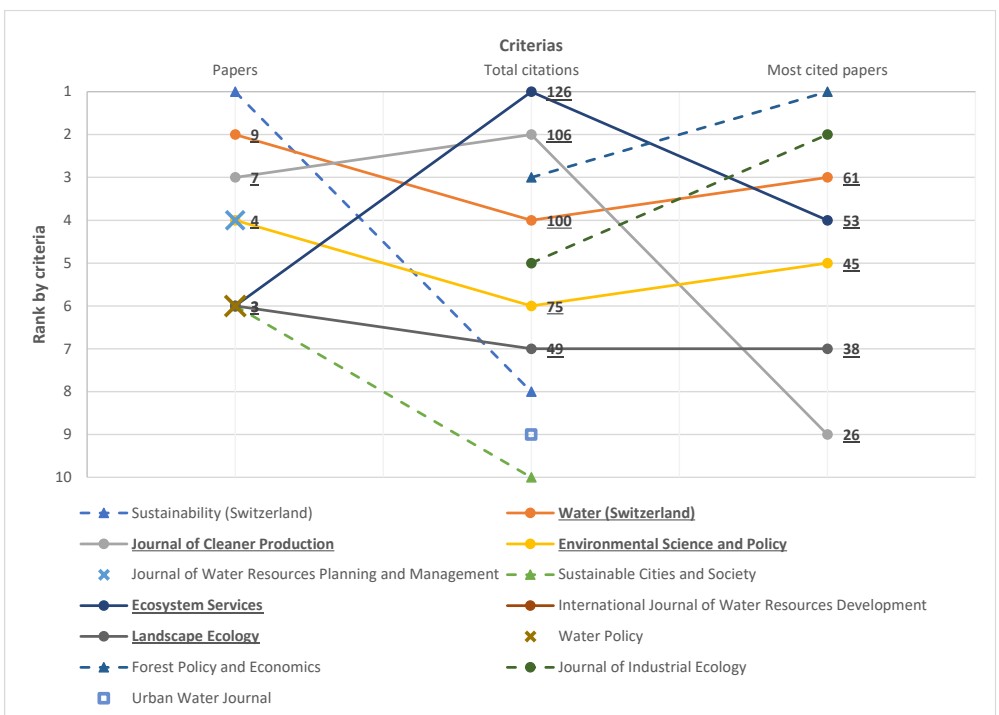

**Figure 6.** Ranking of the selected sample according to the number of documents, citations, and most cited articles.

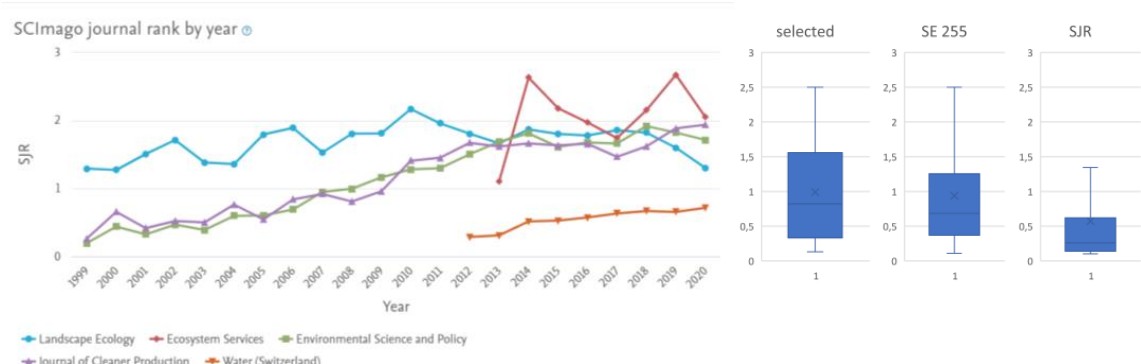

**Figure 7.** Evolution of the SJR index of the main journals in the selected sample (**right**) and box plot graphics of all publications registered in Scimago (SJR) and "water SE" and selected samples (**left**). Adapted from Scopus and SCImago 2020.

### 4.3. Affiliations: Countries and Institutions

The study gathers a sample from 43 countries according to the institutional affiliation of the authors, with Australia, the United States, and the United Kingdom (AUSUK) standing out. According to the Scopus data, this triple accounts for about half of these papers (48.2%, 41 documents), especially by the United States (21).

In addition to the projection of the AUSUK trio, Germany, the Netherlands, and Indonesia show a better capacity for internationalization and propensity for shared scientific development regarding the greater number of co-authorship links (nine and eight, respectively) concerning the number of publications (five, three, and four, respectively). Brazil and Portugal were grouped into clusters led by Germany and the UK, respectively.

According to Ritchie and Roser [36], the lack of access to clean water is especially severe in lower-income countries, with the number of deaths three times higher than homicides and similar to those in traffic accidents around the world. However, on the one

hand, water values involve humanitarian and peace issues in these countries, on the other hand, in developed infrastructure areas the challenge is to maintain these services.

These different realities reflect the relativity of values and can be verified in the sample by comparing the study unit with the water access scenario. The following table presents the number of papers by respective countries studied when presented—except for eight records due to the information absence or the paper approach generality.

Ritchie and Roser [36] also noted that the percentage of deaths attributed to water insecurity in 2019 concentrates on central Africa and south-central Asia. Although the selected articles focused on these continents, they privileged countries with full or wide water access—notably China, South Korea, and South Africa, as underlined on Table 1.

**Table 1.** Total per continent include papers with continental approach.

| Country by Continent | Papers |
| :---: | :---: |
| ASIA | 23 |
| China | 4 |
| India and South Korea | 3 |
| Philippines | 2 |
| Malaysia, Japan, Nepal, Cambodia, Bangladesh, Afghanistan, East Timor, Iran e United Arab Emirates | 1 |
| AMERICA | 24 |
| United States | 10 |
| Canada, Colombia, Brazil e Peru | 2 |
| Nicaragua | 1 |
| Ecuador e Mexico | 1 |
| AFRICA | 15 |
| South Africa | 5 |
| Uganda | 2 |
| Ethiopia, Tanzania, Ghana, Yemen, Kenya e Senegal | 1 |
| EUROPE | 11 |
| Portugal e United Kingdom | 2 |
| Spain, Ireland e Norway | 1 |
| OCEANIA | 4 |
| Australia | 4 |
| **Total** | **77** |

In the same way, to know the institutions involved is a means of understanding the knowledge spread about water value. The nationality of affiliations refers to the institutional data. The study sample was developed by 158 entities, with the Universities of Oxford and Lisbon standing out, with the largest number of publications (4). The International Centre for Forestry Research, the University of Queensland, and nine others appear with 3 and 2 articles, respectively, leaving 145 institutions with only one publication. The involvement of companies, international organizations, and government institutions shows an opportunity for the dissemination and mutual improvement in knowledge.

*4.4. Authors Relationships*

The complexity faced in defining keywords reflects different research approaches in the literature: only 3 (Baral, H.; Mijic, A.; Abdalla, H.) of 315 authors stand out with two publications. They are authors with environmental and social themes in common motivation and are responsible for four articles in the study sample. Baral, H. figures among the 10 most cited articles [37], with 43 citations, and Mijic, A. and Abdalla, H., who write two papers, have in common the institutional link (Imperial College London).

To explore this diversity, the study sample was analyzed by mutual citation. Figure 8 illustrates the co-citation network with ten or more occurrences to emphasize those most recurrent. The texts and nodes sizes indicate the amount of co-authorship, while the lines

thickness means its frequency jointly citation. Only 31 of 315 authors meet this requirement, and 29 with links among themselves grouped in 6 clusters with between 2 and 8 authors.

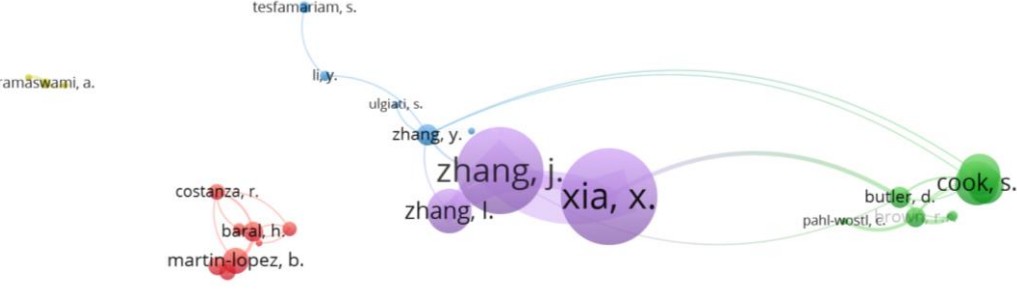

**Figure 8.** Authors' network in the selected sample according to co-citation. Data from Scopus, visualizing by VOSviwer 1.6.18.

From this perspective, five authors wrote 63 of the 4565 referenced papers. The Xia, X. and Zhang, J. general publications focus on the study sample remaining themes (Energy and Engineering). These can be found in 32 and 18 cited documents, respectively, and also stand out as the most frequent pair, as illustrated by the greater thickness of the connection between them, with 10 joint publications. Cook, S. and Sharma, A. further stand out in a specific cluster with 14 and 11 publications consulted each, of which 4 are together.

## 5. Semantic Analysis

The VOSviewer tool allows for the evaluation of the frequency, affinities, and relationships of the authors' keywords, among other possibilities. A similar analysis can be developed by adopting the articles as an analysis unit, suggesting thematic affinities if they share a common knowledge base.

### 5.1. Keywords Network

Figure 9 shows the network of keywords selected by the authors incorporating information as clusters and links, restricted to two or more occurrences, and the formation of groups with four or more elements. The size of the letters and nodes reveals the number of occurrences, and the connections thicken the frequency with which they are flagged together.

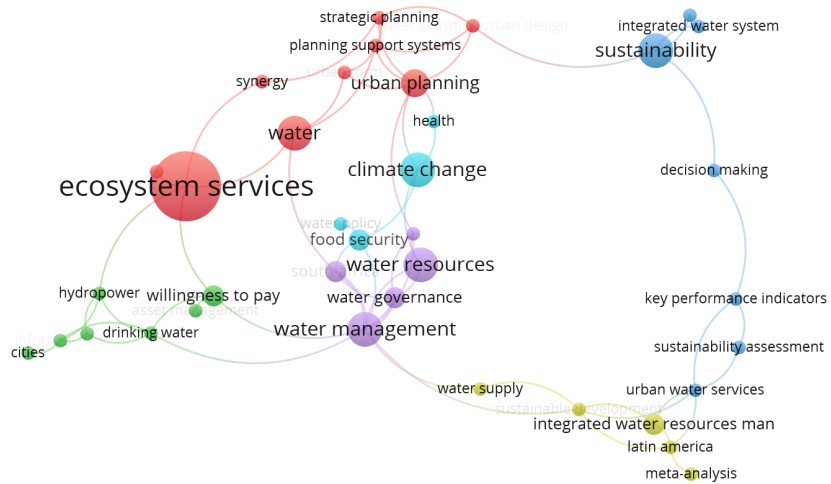

**Figure 9.** Keyword network of the selected sample. Data from Scopus, visualizing by VOSviwer 1.6.18.

In the study, 28 of the 357 keywords meet these requirements and comprise six clusters with between four and nine items. The word "ecosystem service" has the highest number

of occurrences (10), followed by water resources, water management, climate change, and sustainability with five each. The word "water management" presents the highest frequencies of joint citation with "water governance" and "South Africa." This analysis kept all words to preserve the formation of clusters and networks.

The clustering proposed by VOSviewer proposes six groups. In sum, specific clusters comprise individuals' basic needs such as health, food security and resource preservation, positioned in the center of the map and linked to the planning and management themes. The absence of this link with the clusters of keywords related to economic issues (on the left) and decision-making (on the right) suggests the prioritization of other values. This one is integrated with the others by two similar clusters that bring together keywords on sustainability and integration.

### 5.2. Bibliographic Coupling

Figure 10 represents the shared references network, where sizes of the nodes tally the shared references amount, represented by the connection's thickness.

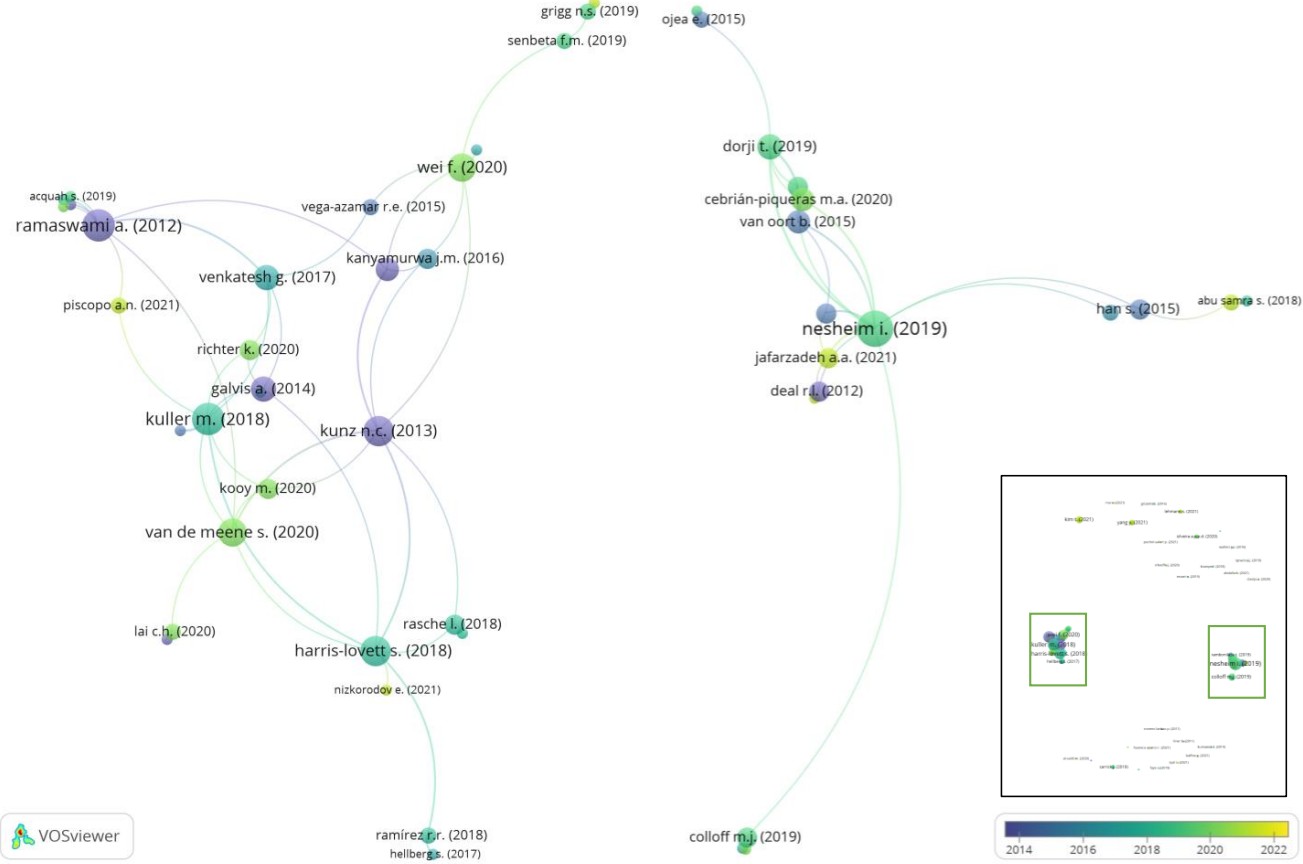

**Figure 10.** Network of selected articles according to reference coupling. Data from Scopus, visualizing adapted from VOSviwer 1.6.18 [37–102].

In the study, 70 of the 84 publications share at least one reference and two networks stand out—analyzed in more detail in a specific subsection below. The in-depth discussion on values is present in a first group of 17 articles (right) that discuss forests, community perceptions, socio-cultural aspects, and "benefits from water and land use functions as important contributions for societal welfare" (Nesheim and Barkved, 2019). These are discussions that point to broad ecosystem or supply values, focusing on users' perceptions as an intervention parameter.

The second group (to the left) gathers 31 selected articles, and it is possible to note that the majority concerned the integrated approach under several aspects. Strongly linked to

the multiple uses of water mentioned above, this context points out new challenges for the segment.

## 6. Narrative Analysis

### 6.1. Values Overview

The Table 2 presents the 14 research motivations identified from the study sample. Five of them (multi-functionalities, user values, economic impacts, service standard and urban resilience) are present in 61.9% (52) publications. As commented, these interests characterize the formation of a network with 18 articles, of which 12 are highlighted below. The multifunctionality that characterizes the integrated approach of the second article cluster also figures prominently in this network.

**Table 2.** Total of papers per research motivation.

| Reseach Motivations | Papers | Total |
|---|---|---|
| Multi-functionality | Wade [38], Tsani et al. [39], Kuller et al. [40], Grizzetti et al. [41], Wei et al. [42], Rezaei et al. [23], Kunz et al. [43], Abu Samra et al. [44], Ramaswami et al. [45], Piscopo et al. [46], Mazumder et al. [6], Liner and Monsabert [47], Lankao [4], Baffoe et al. [48], Colloff et al. [49], Senbeta and Shu [50], Venkatesh et al. [51], Harris-Lovett et al. [52] | 18 |
| User values | Kanyamurwa [53], Ahammad et al. [54], van Oort et al. [37], Doherty et al. [55], Naiga [56], Ma et al. [57], Wanjiru e Xia [58], Cebrián-Piqueras et al. [59], Huaraca et al. [60], Dorji et al. [61], Nesheim e Barkved [62], Han et al. [63], Rambonilaza e Neang [64], Kim et al. [5], Sharma et al. [65] | 15 |
| Economic impact | Teotónio et al. [66], Ojea and Martin-Ortega [67], Ghassemi et al. [68], Jafarzadeh et al. [69], Lai et al. [70], Zhao and Wang [71], Guzmán et al. [72] | 7 |
| Service standard | Rouse [73], Al-Saidi [74], Clavijo et al. [75], Dunford et al. [76], Yang et al. [77], Ignacio et al. [78] | 6 |
| Urban resilience | Barreiro et al. [79], van de Meene et al. [80], Richter et al. [81], Puchol-Salort et al. [82], Wu et al. [83], Nizkorodov [84] | 6 |
| Social | Coleman et al. [85], Lehmann [86], Hellberg [87], Rasche et al. [88], Ramírez and Sañudo-Fontaneda [89] | 5 |
| Environment | O'Keeffe et al. [90], Kuller et al. [40], Galvis et al. [91], Torre et al. [92], Miraji et al. [93] | 5 |
| Food security | Amoah [14], Perez et al. [16], Acquah and Ward [13], Dahik et al. [15] | 4 |
| Financial | Kumawat and Sharma [21], Abdalla et al. [94], Silveira and Mata-Lima [95], Lee et al. [22] | 4 |
| Security | Grigg [96], Carrick et al. [18], Krampe and Gignoux [19] | 3 |
| Poverty | Faye [97], Cronin and Guthrie [98], Jemmali [99] | 3 |
| Health | Gondhalekar et al. [10], Syal [20] | 2 |
| Energy | Binks et al. [11], Bonthuys et al. [12] | 2 |
| Land use | Lieske et al. [17] | 1 |
| n/r | Walters and Chinowsky [100], Deal et al. [101], Vega-Azamar et al. [102] | 3 |

### 6.2. Multiple Use of Water

The multiple uses of water were discussed by 21.4% (18) of the selected articles in the contexts of planning, decision-making, business models, and management. These papers presented in common a broad approach to values and directed the discussions toward multidisciplinary issues.

Social and knowledge development has made the performance of various activities more dependent on infrastructure services, and it has provided a better relationship understanding with natural resources and the inherent outcomes. Kuller et al. [40] extoled the

multiple impacts of water through green infrastructure, such as drinking water quality and flood control. For Harris-Lovett et al. [52], the vital role of wastewater for the recomposition of nutrients in the soil was still little present in water resources management. Venkatesh et al. [51] used two models (WaterMet2—WM2 and Dynamic Metabolism Model—DMM) to quantify externalities under ESS aspects in the mass flows of urban water systems—called urban metabolism.

Grizzetti et al. [41] evaluated the use of River Basin Management Plans (RBMP) in five European countries and noted the difficulties for local actors to understand new concepts and methodologies, but identified the explicitness of indirect values, such as health and multifunctionality, as the main method advantages. Wei et al. [42] cited the social and economic conflicts in various activities and highlighted the benefits of stormwater reuse in case of resource exhaustion to local users.

Ramaswami et al. [45] discussed the importance of infrastructure for the development of urban activities and pollution mitigation. Piscopo et al. [46] emphasized that the contribution of green infrastructure is an important criterion for the process of integrated water management, highlighting the contributions to nutrient recomposition, flood mitigation, and cost reduction. For Lankao [4], the economic objectives prioritized by the neoliberal ideology were incompatible with the environmental, political, and social dimensions of water in view of the prioritization of economic aspects that reflect the values of specific business groups.

### 6.2.1. Users' Perceptions

Users' perceptions were considered by Han et al. [63] as the new frontier in the water sector in order to apply the growing economic approach to infrastructure assets, represented in the sample by the highest research interest (17.9%, 15 papers) in decision-making processes, appropriation of resources, business models, and willingness to pay for service improvement. The authors suggested that service sustainability should consider user satisfaction with other parameters, such as cost reduction. Specifically, they pointed out gaps in objective and subjective values, namely access to data in the first case and satisfaction and feelings related to the service in the second case.

Rambonilaza et al. [64] explored the willingness to pay to encourage more environmentally sustainable production, concluding that this value is higher than that practiced in the local market. Kim et al. [5] inferred that such willingness also resulted from higher satisfaction and a positive perception of service prices. Such values may stimulate new investments or legitimize redistributive pricing policies, and user satisfaction is equally important for the sustainability of community water management in rural Uganda, according to Naiga [56].

Doherty et al. [55] explored the importance of user perception amidst the difficulty of understanding the direct benefits provided by ecosystem protection measures. While Van Oort et al. [37] also discussed the importance of ecosystem services, the results emphasized basic water values: consumption, agriculture, and forest maintenance.

Ahammad et al. [54] faced the relativity of values that results from local socioeconomic conditions, but with the common recognition of the high value of water supply among different income strata. However, the population with lower incomes attributed greater importance to the resource because they consider it essential for health, well-being, and livelihoods. Dorji et al. [61] corroborated this discussion by highlighting the consensus on its importance and vulnerability found among the socio-cultural values of ecosystems. Nesheim et al. [62] proposed a balance in the management of natural resources in view of the subjectivity of the values that benefits assume among governance levels, showing the dependence on context and scale.

### 6.2.2. Economic Impacts

Economic impacts are the third most present topic in the sample, addressed by 8.3% (7) of the selected articles. Ghassemi et al. [68] proposed a model to minimize total costs

from a holistic approach to water consumption and disposal. Teotóno et al. [66] and Guzmán et al. [72] discussed the climate change causalities on energy generation and water supply, finding significant macroeconomic impacts that are mutually related to important intersectoral differences due to the interdependence of water and energy. Ojea and Martin-Ortega [67] acknowledged the advances in the literature on the monetization of water but commented that they were carried out in a fragmented manner and lacked evidence. In contributing to mitigating this gap, they identified factors that systematically influence these values, such as the service type, beneficiaries' profile, method, and context—which reflected the value relativity from these variables.

Jafarzadeh et al. [69] sought to identify crucial local economic benefits through the relationship between land occupation with certain services that develop from the Zagros ecosystem in Iran. The authors highlighted that water has the highest economic value among the other ecosystem services (carbon sequestration, erosion control, and commercial exploitation) and greater synergy with soil preservation, regardless of the type of economic activity. The economic efficiency in water management was addressed by Lai et al. [70] through the system losses, indicating the need to incorporate into public policies the concern with water security, sustainable development and better understanding of the socioeconomic benefits of tariff adjustments. Zhao et al. [71] explored economic methods with environmental ones to evaluate the ecosystem services of Lake Taihu, in China, and found water supply as the main service of an ecosystem among other important for the environment (regulation, protection, and leisure).

### 6.2.3. Level of Services and Urban Resilience

Service levels and urban resilience were addressed by 7.1% (6) articles each. In addition to climate change, anthropological impacts on basic service provision were addressed by Dunfort et al. [76] and Yang et al. [77]. Liner and deMonsabert [47] highlighted the strong economic influence on the detriment of socio-environmental dimensions in decision-making, while Rezaei et al. [23] pointed to little explored values in wastewater recycling. These last two papers suggested models for integrating the three dimensions (ECO, ENV, and SOC) in planning and management.

The role of essential service provision in the resilience of cities is discussed by Barreiro et al. [79] and Meene [80] as infrastructure interdependence and governance complexities, respectively. Richter et al. [81] proposed an integrated sanitation management model (namely water, wastewater and drainage) motivated by flood mitigation and public health risks.

### 6.3. Integrated Approach

The second paper's network by bibliographic coupling presents in common the systemic benefits of water supply as health and food and physical safety. However, the predominance of the integrated approach stems from the discussions in 16 of the 31 articles dealing with planning, resource management, and decision-making, of which some are discussed below.

Grigg [96] presented the various elements involved in water and wastewater service delivery to illustrate features such as versatility, scale, and assets permeability, further highlighting the complexities of implementation, financing, governance, and urban operation. The author explored stormwater systems, pointing out their multiple benefits, but emphasized the need to recognize the values of natural systems and ecosystem services. For Grigg [96], the growing perception of interdisciplinarity stresses the role of water on other types of problems—an effort studied by sector nexus approaches, such as selected works by Teotonio et al. [66], Colloff et al. [49], and Wu et al. [83].

Rouse [73] pointed to data ambiguity on drinking water and wastewater collection and treatment access, developing a case study to address policy challenges and urban supply costs. Questions emerged about the discouragement of pre-existing infrastructure and the need to combine tariff policies with the effectiveness and efficiency of operators. In particular, the author highlighted that the sector success does not depend on the type of

operator (public or private) but on the need to integrate it with city planning to promote adequate governance of long maturation assets. Ramaswami et al. [45] corroborated this analysis by reiterating that the city's sustainability depends on complex interactions between natural systems, shared infrastructure, and governance, highlighting the need for spatial integration.

Other integration issues are a matter of community inclusion through a willingness to pay for improved services [5], coordinated and shared management [88], shared management [50,56], and land use [17,102]. Sharma et al. [65] sought insights into an integrated resource management model (Water Sensitive Urban Design—WSUD), a component of Integrated Urban Water Management (IUWM) in Australia. Wade [38] highlighted the tools available to address the synergy between the three dimensions (ECO, ENV, and SOC) with the emphasis on Integrated Water Resources Management (IWRM) but acknowledged that they lacked greater flexibility to adapt to constant political and territorial change.

## 7. Conclusions

The approach to water values developed in this paper was intended to point out the motivations of academics involving water, such as business performance or its conservation, health, nutrition, and human development. The Scopus tool was used given the best conditions for data analysis and integration, selecting 84 relevant papers published between 2011 and 2020 by analyzing their respective contents. The main limitation of this study is the keyword selection, which cannot cover other interesting studies because of the wider kind of approaches that "value" could assume. Furthermore, the time of analysis (2010 to 2019) and language (English) could be extended according to more resources (time and/or researchers) engaged to this research, and the perception of studies thematic is personnel— so, the papers classification could change by researcher and readers understandings.

The interest in the subject registers a significant increase in publications in this period, especially since 2014 in the social and economic areas given the real growth of these concerning the number of environment papers area. Although those are prioritized, it is possible to perceive the expressive link of remaining studies with this area because of the explicit ecosystem's inferences.

In the study, the papers were published in 47 journals, of which five (Water Switzerland, Journal of Cleaner Production, Environmental Science and Policy, Ecosystem Services and Landscape Ecology) stood out for the greatest number of articles and accumulated citations per article. The water value theme interest can be found in qualified journals according to Scimago, with only one (Landscape Ecology) regressing to the lower classification achieved in their beginner periods.

The water value is mostly discussed by institutions of the AUSUK countries but highlights Germany, Holland, and Indonesia for the greatest international extension. In all, 158 entities were involved, especially the Universities of Oxford and Lisbon for the largest number of publications, as well as the engagement of companies, international organizations, and government institutions, showing the opportunity for dissemination and mutual improvement in knowledge. Such institutional characteristics and heterogeneity approach were also present since only 3 of the 315 researchers stood out with two publications. When evaluated by the amount of co-citation in the sample, only 31 of 315 authors met this requirement, among which 29 were linked by joint publication.

The water value relativity can be noted by the unit analysis: although many studies focused on the basic needs in countries with no or poor drink water access, most were privileged countries with full or wide water access—notably China, South Korea, and South Africa. It is a clue that (1) different values about water coexist, and they (2) need to make clear the contexts to be prioritized. The improvement in water services in many countries increases the attention of often highlighting and maintaining its benefits.

The analysis of the authors' keywords network showed that common values, such as health, food safety, and resource preservation, were linked to planning and management, with no evidence of links to economic and decision-making issues—inferring the

predominance of other, more particular values. The same network approach using articles as the unit of analysis illustrated two predominant clusters formed from the convergence of the bibliographic references used (co-citation). In these terms, one large network gathers articles interested in different water values and another network is more dedicated to an integrated approach under distinct aspects.

Overall, 14 studies' motivations were identified, especially on the resource multifunctionality and the user's perception, corresponding to about 40% of papers, which together with the economic impacts, urban resilience, and service standards, were the main objects of study of a paper network with others 18 articles. The interest in user perception stems from the growing economic approach to infrastructure assets, was an important parameter for the sustainability of these assets—especially the financial ones. The second paper network counted on a greater variety of interests, but they presented in common the concern of integrating these and the water cycle. A general framework for the explicit values at stake, such as local realities, and their relationships, is a way to contribute to a more organic perspective of the resource of policymakers, investor, and society.

**Author Contributions:** Conceptualization, methodology, software, validation, formal analysis, investigation, resources, data curation, writing—original draft preparation, writing—review and editing, supervision, project administration, funding acquisition: J.d.P. and R.M. All authors have read and agreed to the published version of the manuscript.

**Funding:** This research received no external funding.

**Institutional Review Board Statement:** Not applicable.

**Informed Consent Statement:** Not applicable.

**Data Availability Statement:** The "water SEE" sample can be found at https://bit.ly/3mg9F6k (accessed on 6 January 2022).

**Conflicts of Interest:** The authors declare no conflict of interest.

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
