# Peer review of "Water Value Integrated Approach: A Systematic Literature Review"

_water, doi:10.3390/w14121845_

Round 1

Reviewer 1 Report

The article "Water value integrated approach: a systematic literature review" explores a case study on water values in documents. This research experiment involves research approaches by applying semantic keyword analysis and participant perception categorization. The approach of water values developed in this study points out how the authors take the motivations of studies involving water, such as business performance or its conservation, health, nutrition, and human development.

Overall, I have no qualms about the methods presented, however, there is room for improvement by expatiating additional linking statements in terms of water valuing research works and the social background trends. Also, please increase the visibility for the potential readers.

Please take English editing service with a professional proofreader and ask her/him to rephrase the sentences both in results and conclusion sections.

I think this paper is almost ready for publication for now. 

Author Response

We thank the reviewer for his/her comments concerning our manuscript! Due to his/her suggestions, we can present a more appealing paper. We hope the changes meet the expectations.

 The following is our response to the reviewer comments.

I: The article "Water value integrated approach: a systematic literature review" explores a case study on water values in documents. This research experiment involves research approaches by applying semantic keyword analysis and participant perception categorization. The approach of water values developed in this study points out how the authors take the motivations of studies involving water, such as business performance or its conservation, health, nutrition, and human development.

Overall, I have no qualms about the methods presented, however, there is room for improvement by expatiating additional linking statements in terms of water valuing research works and the social background trends. Also, please increase the visibility for the potential readers.

Please take English editing service with a professional proofreader and ask her/him to rephrase the sentences both in results and conclusion sections.

I think this paper is almost ready for publication for now.

Response:

We would like to thank you for the comments  that contribute to a better theme discussion and understanding. We improved some passages to enhance the links with the water valuing research, as following:

"The water value relativity can be noted by the unit analysis: although many studies focus on basic need in countries with no or poor drink water access, most of then privilege countries with full or wide water access - notably China, South Korea, and South Africa. It is a clue that 1) different values about water coexist, those 2) need to make clear the contexts to be prioritized. The improvement of water services in many countries increases the attention to often highlight and maintain its benefits." (Included in Conclusion section, see lines 574-579).

It is important to be noted that the effort expended was to keep a general view, not focusing in social, economic, or environmental fields. So, these words were avoided to keep this general discussion, such as lines 138-141, 153 and 157, and so on.

To seal the article, the potential readers were cited in the last paragraph (lines 595-597)

"A general framework for explicit the values at stake such as local realities, and its relationships, is a way to contribute to a more organic perspective of the resource of policy makers, investor, and society."

Thanks for your suggestions and recommendations! They helped us improve the paper substantially.

Reviewer 2 Report

This paper deals with an interesting subject of water value. It is based on an extensive literature review. However, the lecture of the paper is laborious. The authors have to make the following major revision.

1)    The introduction

The objective and novelty of the paper should be better addressed.

2)    Section “Basic and relative values”

This section should include a discussion of the basic and relative values in the water domain.

3)    Section method

Add a diagram that summarizes the methodology.

4)    Section Quantitative review 

This section is long. It includes some subsections which are not directly related to the paper subject “water value”: Journals overview, Affiliations: countries and institutions, and Authors relationships. These sections could be deleted. If the authors want to keep them, they have to add some discussion about their contribution to the paper subject.

5)    Conclusion

The conclusion should focus on the main results concerning the paper's subject and discuss the limitation of this research.

Author Response

We thank the reviewer for his/her comments concerning our manuscript! Due to his/her suggestions, we can present a more appealing paper. We hope the changes meet the expectations.

The following is our response to the reviewer comments.

I: This paper deals with an interesting subject of water value. It is based on an extensive literature review. However, the lecture of the paper is laborious. The authors have to make the following major revision.

1)    The introduction

The objective and novelty of the paper should be better addressed.

Response:

We appreciate the objectivity of your comments those contribute to a better theme discussion and comprehension. We did a review those include paragraph development and structure check, as pointed above:

"This article develops a systematic literature review to identify the motivations of related studies and then highlight the main water values focused by the academics, such as business performance or its conservation, health, nutrition, and human development. Develops an open methodology using bibliometric data and visual analysis those suggest thematic affinities, point to key institutions and countries interested in this theme, and uses a comprehensive presentation to deal with the sectorial characteristic of multi-level and -disciplinary stakeholders." (Lines 84-90).

2)    Section “Basic and relative values”

This section should include a discussion of the basic and relative values in the water domain.

Response:

Thanks for pointing out this matter. This is a hard and controversy discussion, reason why we appeal to the psychology and social area concepts for the highlight main concepts. Nevertheless, we strive to keep the water value discussion alive, such as the following passages:

"Gondhalekar et al. [10] expose the echoes of the collective and private water values clashes by ratifying, respectively, the direct relationship of diarrhea with access to safe drinking water and assigning some share of blame to tourism for the pressure of fluctuating demand in Leh District, India. Water is also responsible for life quality through power generation [11,12], agricultural production [13–16], land development [17], security [18] and mitigation of social conflicts [19,20]. They also related it to the satisfaction of specific groups that seek business development through higher enterprises’ performance [6,21–23]. Such motivations, or goals, expose the predominant value attributed to water - be it a business, human survival, or environmental conservation." (lines 112-120).

"These contrasts are clear in the preservation of resources and dominion (or even the "backyards") to detriment of diffuse benefits, reproducing conflicts between individual and collective values." (lines 134-136).

3)    Section method

Add a diagram that summarizes the methodology.

Response:

Thanks for your remarks because diagrams are a good way to summarize and improve the research logical (line 185).

4)    Section Quantitative review

This section is long. It includes some subsections which are not directly related to the paper subject “water value”: Journals overview, Affiliations: countries and institutions, and Authors relationships. These sections could be deleted. If the authors want to keep them, they have to add some discussion about their contribution to the paper subject.

Response:

Definitely, a data description is a difficult task to keep the readers interest - but these subsections are important to show a general literature data about the theme. To increase the readability, some passages and figures was excluded, and the follow phrase was added:

"They have common interest in environment science thought themes such as production, policies, and management. Water is the specific focus of one of then, covered by 13 research areas, as well as the economic and social values of ecosystem services." (lines 277-280).

5)    Conclusion

The conclusion should focus on the main results concerning the paper's subject and discuss the limitation of this research.

Response:

We want to thank you for your suggestion because the conclusion is an important part of the paper. A section review was done, and some passages added, namely:

"The water value relativity can be noted by the unit analysis: although many studies focus on basic need in countries with no or poor drink water access, most of then privilege countries with full or wide water access - notably China, South Korea, and South Africa. It is a clue that 1) different values about water coexist, those 2) need to make clear the contexts to be prioritized. The improvement of water services in many countries increases the attention to often highlight and maintain its benefits." (lines 574-579).

"A general framework for explicit the values at stake such as local realities, and its relationships, is a way to contribute to a more organic perspective of the resource of policy makers, investor, and society." (lines 595-597).

Likewise, sum up the studies limitations are also important to delimit the results and conclusions, and highlight ways to future improvements. Thank you for point this missing.

"The main limitation of this study is the keyword selection, which cannot cover other interesting studies because the wider kind of approaches that “value” could assume. Furthermore, the time of analysis (2010 to 2019) and language (English) could be extended according to resources (time and/or researchers) engaged to this research, and the perception of studies thematic is personnel – so, the papers classification could change by researcher and readers understandings." (lines 549-553).

Thanks for your suggestions and recommendations! They helped us improve the paper substantially.

Round 2

Reviewer 2 Report

The authors addressed well my comments. The p